# Saliency Detection Based on the Combination of High-Level Knowledge and Low-Level Cues in Foggy Images

**DOI:** 10.3390/e21040374

**Published:** 2019-04-06

**Authors:** Xin Zhu, Xin Xu, Nan Mu

**Affiliations:** 1School of Computer Science and Technology, Wuhan University of Science and Technology, Wuhan 430065, China; 2Hubei Province Key Laboratory of Intelligent Information Processing and Real-time Industrial System, Wuhan 430065, China; 3School of Electronic Information and Electrical Engineering, Shanghai Jiao Tong University, Shanghai 200240, China

**Keywords:** saliency detection, foggy image, spatial domain, frequency domain, object contour detection, discrete stationary wavelet transform

## Abstract

A key issue in saliency detection of the foggy images in the wild for human tracking is how to effectively define the less obvious salient objects, and the leading cause is that the contrast and resolution is reduced by the light scattering through fog particles. In this paper, to suppress the interference of the fog and acquire boundaries of salient objects more precisely, we present a novel saliency detection method for human tracking in the wild. In our method, a combination of object contour detection and salient object detection is introduced. The proposed model can not only maintain the object edge more precisely via object contour detection, but also ensure the integrity of salient objects, and finally obtain accurate saliency maps of objects. Firstly, the input image is transformed into HSV color space, and the amplitude spectrum (AS) of each color channel is adjusted to obtain the frequency domain (FD) saliency map. Then, the contrast of the local-global superpixel is calculated, and the saliency map of the spatial domain (SD) is obtained. We use Discrete Stationary Wavelet Transform (DSWT) to fuse the cues of the FD and SD. Finally, a fully convolutional encoder–decoder model is utilized to refine the contour of the salient objects. Experimental results demonstrate that the presented model can remove the influence of fog efficiently, and the performance is better than 16 state-of-the-art saliency models.

## 1. Introduction

There is great influence on the visibility of the human tracking in the wild under foggy environments on account of how dust particles suspend in the air. Therefore, the foggy images typically have low contrast and faded color features, in which the main objects are difficult to be recognized. Saliency detection is advantageous to this task, and it is a cognitive process that simulates the attention mechanism of human visual system (HVS) [1,2,3], which has an astonishing capability to rapidly judge the most attractive image region from a scene for further processing in the human brain.

In the past several years, the detection of visual salient objects has drawn much attention to most image processing applications. Saliency detection in foggy images acts a pivotal part in fields such as human tracking in the wild, object recognition, object segmentation, remote sensing, intelligent vehicles, and surveillance. So far, all kinds of defogging techniques [4,5,6,7] have been proposed, and they can reach comparatively good performance.

At present, image processing methods in foggy weather can be split into image enhancement and image restoration methods.

Image restoration methods include Dark channel prior algorithm [8], Visual enhancement algorithms for uniform and non-uniform fog [9], and defogging algorithms based on deep learning [10]. The method of image restoration based on the physical model is mainly to explore the physical mechanism of images degraded by fog, and to establish a general foggy weather degradation model. Then, the degradation model is calculated to compensate for the loss of image information caused by the degradation process. Finally, the quality of foggy images can be improved. However, the image restoration algorithm is a physical model based on atmospheric scattering. It requires more priori knowledge. Image enhancement methods can be divided into contrast enhancement and color enhancement. Image enhancement representative algorithms include histogram equalization [11], Retinex [12], and Wavelet based approaches [13,14]. However, the main drawbacks of these algorithms include: (1) High complexity makes their execution time-consuming, thereby making it difficult to guarantee the real-time performance of saliency detection. (2) During the process of dehazing, the visibility of foreground and background is increased simultaneously, so the recognition of salient objects is disturbed to some extent. (3) Image color distortion leads to visual features such as the edge and contour of the target cannot be accurately extracted.

Due to the low-resolution and low-contrast characteristics of foggy images, traditional spatial or frequency-based saliency models have a poor performance under fog environment. In view of this problem, this paper presents a frequency-spatial saliency model based on the atmospheric scattering distribution of foggy images, which can obtain effective information under foggy weather. Since the traditional machine learning method leads to the loss of boundary information, the object contour detection method of deep learning is added to enrich the edge information of the saliency map. As illustrated in Figure 1, the object contour detection method obviously improves the quality of the saliency map.

In this paper, traditional methods and deep learning methods are combined to effectively detect salient objects for human tracking in the wild. In step one, the frequency domain (FD) and the spatial information are fused by DSWT. We utilize the object contour detection method of deep learning to obtain the map of the edge of the object at step two. Last, we obtain the final saliency map of the foggy image by fusing the two maps. Specifically, in step one, the foggy image is transformed into HSV color space first, and the amplitude information of FD is utilized to obtain feature maps in each channel. Then, segmenting the image into superpixels and computing the saliency of each superpixel by the local-global spatial contrast. Finally, the DSWT is applied to fuse the FD and spatial domain (SD) saliency maps, and the Gaussian filter is employed to refine the results. The flow diagram of the presented method is shown in Figure 2. The experimental results show that the proposed method can effectively detect salient objects under fog conditions.

## 2. Related Works

Saliency detection is generally driven by low-level knowledge and high-level cues. Therefore, visual saliency computation under foggy environments for human tracking in the wild can be typically categorized into two classes: Saliency computational models and object contour detection approaches. Traditional saliency computational models are data-driven and primarily utilize low-level image features; while top-down object contour detection models are task-driven and usually utilize cognitive visual features.

### 2.1. Saliency Computational Models

From the perspective of information processing, traditional saliency models can be divided into two categories: SD and FD based models.

The SD saliency models are usually based on the contrast analysis to establish the algorithms. Itti et al. [15] presented a famous saliency model by utilizing the center-surround differences of multiple features. Goferman et al. [16] introduced a context-aware saliency approach, which measures the similarity of image patches in a local-global manner. Xu et al. [17] proposed a superpixel-level saliency method through a support vector machine (SVM) to train unique features. Cheng et al. [18] considered the histogram information and spatial relations, and then developed a global contrast-based saliency algorithm. Peng et al. [19] integrated tree-structed sparsity-inducing and Laplacian regularizations to construct a structured matrix decomposition model. However, most of the features used in these spatial models are not ideal for foggy images.

The saliency models of the FD develop an algorithm by converting the to a spectrum. Hou and Zhang [20] employed a spectral residual saliency method, which utilizes the log spectra to represent images. Guo et al. [21] extended the FPT algorithm and denoted four features of image by quaternion. Then, they utilized the Fourier transform of the quaternion to acquire the saliency map. Achanta et al. [22] built a frequency-tuned method, which estimates the contrast of several features. The color and brightness characteristics of each pixel are adopted to calculate the saliency map by Bian and Zhang [23]. Li et al. [24] explored saliency detection by analyzing the scale-space information of the amplitude spectrum (AS). Li et al. [25] studied the image saliency in the FD to design the model. Arya et al. [26] integrated local and global features to propose a biologically feasible FD saliency algorithm. These existing FD saliency models do not work well in foggy images due to the low-frequency information representing salient objects are greatly reduced in foggy weather.

### 2.2. Object Contour Detection

Object contour detection is a traditional computer vision problem with a long history. The traditional computer vision methods include Roberts, Prewitt, Sobel, canny, and other algorithms.

In the process of object contour detection, Roberts’ algorithm does not smooth the image, so the image noise is generally not well suppressed, which also affects the loss of a part of the edge when calculating the positioning. However, Roberts’ algorithm has higher positioning accuracy and better effect on steep low-noise images. Prewitt algorithm can suppress noise. The principle of noise suppression is pixel average, which is equal to low-pass filtering of the image. Thus, Prewitt’s algorithm is inferior to Roberts’ algorithm in edge positioning. The practical application of the Sobel edge detection algorithm [27] is when the efficiency requirements are high and the fine texture is not of interest. Sobel is usually directional and can detect only vertical or horizontal edges or both. The Sobel algorithm is improved on the basis of the Prewitt algorithm. Compared with the Prewitt algorithm, the Sobel algorithm can suppress the smoothing noise better. The Canny algorithm [28] pays more attention to the edge information reflected by the pixel gradient change and does not consider the actual object. However, it leads to loss of spatial information of the image at the same time. For some images where the edge color is similar to the background color, the edge information may be lost. The Canny algorithm is one of the best algorithms for detecting edge effects in traditional first-order differentials. It has stronger denoising capabilities than the Prewitt and Sobel algorithms. On the other hand, it is also easy to smooth some edge information, and its checking method is more complicated. However, the traditional edge detection algorithm uses the maximum gradient or the zero-crossing value of the second derivative to obtain the edge of the image. Although these algorithms have better real-time performance, they have poor anti-interference and cannot effectively overcome the influence of noise. In addition, the positioning is not good.

With the development of deep learning, the fast edge algorithm, HED and RCF algorithms are introduced. Fast edge algorithm [29] uses random forests to generate edge information. Ground truth is used to extract the edge of the image patch. This can not only reflect the actual object, but also reflect the spatial information of the picture. HED [30] used the network modified by VGG. Feature information is extracted from the whole image through multi-scale fusion, multi-loss and other methods. Similarly, it can reflect the feature information of the edge. RCF [31] takes advantage of the features of all convolutional layers in each stage compared to HED. The use of more features has also brought about an improvement in results and achieved good results. Inspired but different from these deep learning models, we employ an encoder-decoder network with full convolution to guide better salient object detection.

In our previous work, we trained an encoder-decoder network with full convolution using Caffe to optimize the performance of saliency detection. The proposed fully convolutional encoder-decoder network can learn the object contour to better represent saliency map in low contrast foggy images. The key contributions of this paper are summarized below: (1) We compute the saliency map via a frequency-spatial fusion saliency model based on DSWT. (2) This framework is further refined by a fully convolutional encoder-decoder model based on fully convolutional networks [32] and deconvolutional networks [33]. (3) The presented saliency computational model has better performance in foggy images than traditional models.

## 3. Proposed Saliency Detection Method

In this paper, we propose a frequency and spatial cues based traditional method through DSWT and a deep learning-based edge detection method fused salient object computational model to obtain the saliency map in foggy images effectively.

This section first analyses the features of foggy images, including the imaging model and effect of fog distortion on images in Section 3.1. We describe the FD based algorithm and some important computational formulas in Section 3.2. Then we give the detailed description of the SD based algorithm in Section 3.3. Section 3.4 provides the implementation of the discrete stationary wavelet transform based image fusion, which combines the above-mentioned two algorithms to generate elementary saliency map. Finally, Section 3.5 introduces the object contour detection method to refine the contour of the saliency map. It makes the position of the salient object more precise.

### 3.1. Analysis of Foggy Image Features

#### 3.1.1. Imaging Model of Foggy Image

Under fog conditions, there are a lot of tiny water droplets and aerosols in the atmosphere, which seriously affect the spread of light, resulting in a decrease in image clarity and contrast in foggy days. Especially for color images, it also produces severe color distortion and misalignment. From the respective of the computer vision, there are plentiful models [34,35] which are widely used for describing the information of foggy images. Narasimhan and Nayar [35] proposed imaging model of foggy images as shown following:(1)$$I_{x}^{c}=J_{x}^{c}t_{x}+A^{c}(1-t_{x})$$
where c∈{r, g, b} denotes the color space of the images and $I_{x}^{c}$ denotes the foggy image captured by an imaging device. $J_{x}^{c}$ and $t_{x}$ denote the scene reflected light and scene transmissivity, respectively. $A^{c}$ is a constant and represents the ambient light.

In Equation (1), $J_{x}^{c}t_{x}$ and $A^{c}(1-t_{x})$ denote the direct attenuation [10] and air light [36], respectively. Direct attenuation is defined as the radiance of the scene and its attenuation in the medium. Air light, on the other hand, is caused by the previous scattering light, resulting in a change in the color of the scene. The transmission t can be indicated as follows in which the atmosphere is homogenous:(2)$$t_{x}=e^{-\beta d_{x}}$$
let β denote the scattering coefficient of the atmosphere.

The results show that the scene brightness decays exponentially with the scene depth *d*.

#### 3.1.2. Effect of Foggy Distortion on Images

The degraded effect of fog on the image [37] is called fog distortion. The degraded effect of fog distortion brings great challenges to the saliency computation of images. The effect of fog distortion on image quality is mainly concentrated in three aspects:
(1)The original information of the image is destroyed by the fog, and the structural information of the image is regarded as the high frequency component with enough energy in the image. The generation of fog destroys the structural information of the image and affects the details and texture of the scene object.(2)The fog adds some information to the image. The existence of fog can be seen as adding the relevant channel information of the image and making the overall brightness of the image rises.(3)Fog distortion combines with the original information of the image to generate some information. Due to the interaction between fog particles and the information of the image itself, the foggy image adds some multiplicative information, such as fog noise. It blurs the image, which reduces the contrast of the image.

### 3.2. FD Based Algorithm

Given a foggy image, it is transformed into HSV color space firstly, which has shown strong stimuli to human visual cortex in foggy image [38], thus the hue, saturation, value features of H, S, and V channels can be considered as the important indicators for detecting saliency.

Then, the H, S, and V channels are converted into FD respectively by conducting the Fast Fourier Transform (FFT) as:(3)$$F(u,v)=\sum_{x=0}^{M-1}\sum_{y=0}^{N-1}f(x,y)e^{-j2\pi \left(\frac{ux}{M}+\frac{vy}{N}\right)},$$
where M and N denote the image’s width and height. f(x,y) and F(u,v) denote image pixels in SD and FD, respectively.

A(u,v) and P(u,v) represent the AS and the *phase spectrum* (PS), respectively. And they can be computed via:(4)P(u,v)=angle(F(u,v)),
(5)A(u,v)=abs(F(u,v)),
where the AS function and the PS function are denoted as abs(⋅) and angle(⋅), respectively. In PS function, each element of the complex array F(u,v) returns the phase angle (in radians). This angle is between ±π. Amplitude spectrum A(u,v)=abs(F(u,v)) means the absolute value of image pixels in frequency domain.

For foggy images, the low amplitude in FD can be regarded as a cue of the object, and the high amplitude can represent the fog background. Therefore, restraining the high amplitude information to highlight the object region in other words, the salient object can be extracted by removing the peaks of the AS via:(6)A(u,v)=medfilt2(A(u,v)),
where the median filter function is represented as medfilt2(⋅), which can effectively eliminate the peaks of A(u,v). medfilt2(I) performs median filtering of the image I in two dimensions. Each output pixel contains the median value in a 3-by-3 neighborhood around the corresponding pixel in the input image.

Next, it can compute a new FD map via:(7)$$F(u,v)=\left|A(u,v)\right|e^{-jP(u,v)},$$
where the absolute value is represented as |⋅|.

The FD map is then transformed back to SD by performing the Inverse Fast Fourier Transform (IFFT) via:(8)$$f(x,y)=\frac{1}{MN}\sum_{u=0}^{M-1}\sum_{v=0}^{N-1}F(u,v)e^{j2\pi \left(\frac{ux}{M}+\frac{vy}{N}\right)}.$$

The saliency maps (denoted as Hmap, Smap, and Vmap) of each channel in HSV color space can be acquired by (3)–(8).

Finally, we calculate the sum of Hmap, Smap, and Vmap, and obtain the map of FD saliency (represented as S1).

### 3.3. SD Based Algorithm

To reduce the amount of computation and guarantee the integrity of the object, the input foggy image is first divided into superpixels (presented as SP(i), i=1,⋅⋅⋅,Num, Num=300) through the simple linear iterative clustering (SLIC) algorithm [39]. Then, the obtained $H_{map}$, $S_{map}$, and $V_{map}$ of H, S, and V channels are regarded as the features of saliency.

The local-global saliency of every superpixel SP(i) in $H_{map}$ can be obtained through:(9)$$S_{H_{map}}(i)=1-exp\left\{-\frac{1}{Num-1}\sum_{j=1,j\neq i}^{Num}\frac{d_{H_{map}}(SP(i),SP(j))}{1+E(SP(i),SP(j))}\right\},$$
where $d_{H_{map}}(SP(i),SP(j))$ is the difference in the mean of SP(i) and SP(j) in $H_{map}$. The mean Euclidean distance between SP(i) and SP(j) is represented as E(SP(i),SP(j)).

Through (9), saliency values $S_{S_{map}}(i)$ and $S_{V_{map}}(i)$ of superpixels SP(i) in $S_{map}$ and $V_{map}$ can be figured out.

In the end, the saliency value of each pixel SP(i) is acquired by the sum of $S_{H_{map}}(i)$, $S_{S_{map}}(i)$, and $S_{S_{map}}(i)$. And $S_{2}$ is the saliency map of SD.

### 3.4. DSWT Based Image Fusion

The presented model mainly employs 2-levels DSWT to remove the noise of the saliency map and to accomplish the wavelet decomposition on it.

Low-pass filter and high-pass filter of the 1-level conversion are represented as $h_{1}[n]$ and $g_{1}[n]$. Up sample of the 1-level can calculate the 2-levels filters $h_{2}[n]$ and $g_{2}[n]$. Next, we can obtain the horizontal high-frequency subband $H_{2}$, the approximation low-pass subband $A_{2}$, and the diagonal high-frequency subband $D_{2}$, the vertical high-frequency subband $V_{2}$. The high-pass and low-pass subband has the same size as the initial image. Therefore, the information of detail can be preserved adequately. Thereby, it makes DSWT have translation invariance.

According to above steps, the saliency map based on the FD $S_{1}$ and the saliency map based on the SD $S_{2}$ is obtained. Then, we fuse the two maps through the 2-levels DSWT as:(10)$$\left[A_{1}S_{1},\;H_{1}S_{1},\;V_{1}S_{1},\;D_{1}S_{1}\right]=\mathrm{swt}2(S_{1},\;1,\;‘\mathrm{sym}2’),$$
(11)$$\left[A_{1}S_{2},\;H_{1}S_{2},\;V_{1}S_{2},\;D_{1}S_{2}\right]=\mathrm{swt}2(S_{2},\;1,\;‘\mathrm{sym}2’),$$
(12)$$\left[A_{2}S_{1},\;H_{2}S_{1},\;V_{2}S_{1},\;D_{2}S_{1}\right]=\mathrm{swt}2(A_{1}S_{1},\;1,\;‘\mathrm{sym}2’),$$
(13)$$\left[A_{2}S_{2},H_{2}S_{2},V_{2}S_{2},D_{2}S_{2}\right]=\mathrm{swt}2(A_{1}S_{2},\;1,\;‘\mathrm{sym}2’),$$
where the multilevel DSWT is represented as swt2(⋅). swt2(⋅) performs a multilevel 2-D stationary wavelet decomposition using either an orthogonal or a biorthogonal wavelet. Equations (10)–(13) compute the stationary wavelet decomposition of the real-valued 2-D or 3-D matrix at 1-level by using ‘sym2′. The output three-dimensional array $A_{i}S_{j}$ is represented as the result of the i-level low frequency approximation coefficients of saliency map $S_{j}$ employing ‘sym2’ filter, and $D_{i}S_{j}$, $H_{i}S_{j}$, $V_{i}S_{j}$ represent the high frequency coefficients of the diagonal, vertical and horizontal directions, respectively.

Next, the 2-level fusion is calculated using the following formulas:(14)$$A_{2}S_{f}=0.5\times \left(A_{2}S_{1}+A_{2}S_{2}\right),$$
(15)$$H_{2}S_{f}=D\cdot H_{2}S_{1}+\tilde{D}\cdot H_{2}S_{2},\;D=\left(\left|H_{2}S_{1}\right|-\left|H_{2}S_{2}\right|\right)\geq 0,$$
(16)$$V_{2}S_{f}=D\cdot V_{2}S_{1}+\tilde{D}\cdot V_{2}S_{2},\;D=\left(\left|V_{2}S_{1}\right|-\left|V_{2}S_{2}\right|\right)\geq 0,$$
(17)$$D_{2}S_{f}=D\cdot D_{2}S_{1}+\tilde{D}\cdot D_{2}S_{2},\;D=\left(\left|D_{2}S_{1}\right|-\left|D_{2}S_{2}\right|\right)\geq 0.$$

The 1-level fusion is calculated using the following formulas:(18)$$A_{1}S_{f}=\mathrm{iswt}2(A_{2}S_{f},\;H_{2}S_{f},\;V_{2}S_{f},\;D_{2}S_{f},\;‘\mathrm{sym}2’),$$
(19)$$H_{1}S_{f}=D\cdot H_{1}S_{1}+\tilde{D}\cdot H_{1}S_{2},\;D=\left(\left|H_{1}S_{1}\right|-\left|H_{1}S_{2}\right|\right)\geq 0,$$
(20)$$V_{1}S_{f}=D\cdot V_{1}S_{1}+\tilde{D}\cdot V_{1}S_{2},\;D=\left(\left|V_{1}S_{1}\right|-\left|V_{1}S_{2}\right|\right)\geq 0,$$
(21)$$D_{1}S_{f}=D\cdot D_{1}S_{1}+\tilde{D}\cdot D_{1}S_{2},\;D=\left(\left|D_{1}S_{1}\right|-\left|D_{1}S_{2}\right|\right)\geq 0,$$
where the inverse DSWT function is represented as iswt2(⋅). For example, X=iswt2(A,H,V,D,‘sym2’) reconstructs the matrix X based on the multilevel stationary wavelet decomposition structure [A,H,V,D] in Equation (18) and Equation (22).

Then, the fusion image can be calculated using the following formulas:(22)$$Salmap=\mathrm{iswt}2\left(A_{1}S_{f},H_{1}S_{f},V_{1}S_{f},D_{1}S_{f},‘\mathrm{sym}2’\right).$$

In the end, the proposed method utilizes a Gaussian filter to generate a smoothed saliency map.

### 3.5. Object Contour Detection

Object contour detection model [40] can filter and ignore the edge information in the background and obtain the contour detection result by centering the object in the foreground. Inspired by the fully convolutional networks and deconvolutional networks [33], an object contour detection model is introduced to extract the target contour and suppress background boundaries.

The layers up to ‘fc6′ from VGG-16 [41] are used in the edge detection model as the encoder of the network. The deconv6 decoder convolutional layer uses 1×1 kernel, and all remaining decoder convolutional layers use 5×5 kernel. Except for the decoder convolutional layer next to the output layer which uses the sigmoid activation function, all other decoder convolutional layers are followed by the relu activation function.

We trained the network using Caffe. The parameters of the encoder are fixed when training the network, while only the parameters of the decoder are optimized. This maintains the generalization of the ability of the encoder and enables the decoder network to be easily combined with other tasks.

## 4. Experimental Results

### 4.1. Experiment Setup

**Datasets:** Abundant experiments are executed on two datasets to assess the performance of the proposed saliency model.

A foggy image dataset (FI) was collected from the Internet, which contained 200 foggy images. We also provide the corresponding manual labeled ground truths. The FI dataset can be downloaded at https://drive.google.com/file/d/1aqro3U2lU8iRylyfJP1WRKxTWrrFzizh/view?usp=sharing. The other one is the BSDS500 Dataset. It includes 500 natural images with carefully annotated boundaries by different users. The dataset is divided into three parts: 200 for training, 100 for validation and the other 200 for testing. Object contour detection is utilized to optimize the saliency map which was obtained by traditional machine learning methods of salient object detection. Due to the use of traditional methods, the edge information of the saliency map is incomplete.

**Evaluation Criteria:** For quantitative evaluation, the average computation time, the mean absolute error (MAE) score, the overlapping ratio (OR) score, the precision-recall (PR) curve, the true positive rates (TPRs) *and false positive rates* (FPRs) curve, the area under the curve (AUC) score, the F-measure curve, the weighted F-measure (WF) score, and various saliency models are computed, respectively.

The precision, recall, TPR and FPR values are generated by converting the saliency map into binary map via thresholding to compare the difference of each pixel with ground truth. $\beta^{2}$ is the parameter to weigh the precision and recall, which is set to 0.3 in our experiments [18,22].

The ratio of the number of salient pixels correctly labeled to all salient pixels in this binary map is defined as the precision. In other words, precision refers to how many of the samples that are positively judged by the model that are true positive samples. The recall rate refers to how many positive samples are judged as positive samples by the model in the ground-truth map:(23)$$precision=\frac{\left|TS\cap DS\right|}{\left|DS\right|},recall=\frac{\left|TS\cap DS\right|}{\left|TS\right|},$$
where *TS* and *DS* denote true salient pixels and detected salient pixels by the binary map, respectively.

The TPRs represents the probability that have a right classification of positive examples, and the FPRs represents the probability of splitting a negative sample into a positive sample.
(24)$$\mathrm{TPR}=\frac{TP}{\left(TP+FN\right)},\mathrm{FPR}=\frac{FP}{\left(FP+TN\right)},$$

F-measure value, denoted as $F_{\beta }$, is obtained by computing the weighted harmonic mean of precision and recall.
(25)$$F_{\beta }=\frac{(1+\beta^{2})\times \mathrm{Precision}\times \mathrm{Recall}}{\beta^{2}\times \mathrm{Precision}+\mathrm{Recall}},$$
where $\beta^{2}$ is set to 0.3 to weight precision more than recall as suggested in [42].

Given a ground truth main subject region G and a detected main-subject region D. The OR score is the ratio between two times the correctly detected main-subject region to the sum of detected and ground truth main subject region.
(26)$$\mathrm{OR}=\frac{2\times A(D\cap G)}{A(D)+A(G)},$$

The percentage of area under the TPRs-FPRs curve is called as the AUC score. It intuitively reflects the classification ability of ROC curve.
(27)$$\mathrm{AUC}=\frac{\sum_{i\in positiveClass}rank_{i}-\frac{M(1+M)}{2}}{M\times N},$$

The MAE score to calculate the average difference of each pixel between the saliency map which is predicted and ground truth. It is acquired by:(28)$$\mathrm{MAE}=\frac{1}{W\times H}\sum_{x-1}^{W}\sum_{y-1}^{H}\left|S\left(x,y\right)-G\left(x,y\right)\right|,$$
where *S* is predicted saliency map and *G* is ground truth, the width and height of saliency map *S* are presented as *W* and *H*.

### 4.2. Comparison and Analysis

The presented method is compared with 16 well-known saliency detection methods including: IT [15], CA [16], SMD [19], SR [20], FT [22], MR [41], NP [43], IS [44], LR [45], PD [46], SO [47], BSCA [48], BL [49], GP [50], SC [51], and MIL [52]. The source code provided by others was used to test on our foggy dataset. Each foggy image in our dataset was tested on 16 methods of others to produce the corresponding saliency map.

Figure 3 shows the PR, TPRs-FPRs, and F-measure curves of various saliency models to evaluate the proposed model quantitatively. The larger the area under the curve is, the better the performance of the saliency model will be.

It can be seen from the three figures that the proposed model is superior to other saliency models, which validates that our saliency result is robust in foggy images.

The greatest three results in Table 1 are emphasized in red fonts, blue fonts and green fonts when comparing performance with other methods. Table 1 shows that the presented model yields the greatest performance in terms of AUC and OR scores and obtains the second best in MAE and WF. These results indicate that the presented saliency model reaches the better performance under fog conditions. Moreover, our proposed method has a shorter running time than most, ranking fifth out of other 16 methods.

Figure 4 shows the visual comparisons of varieties of saliency detection models on the foggy image dataset, which demonstrates that the saliency maps obtained by our method are much closer to the ground truths. Compared to the baselines, our method yields a better performance, which means that it suppresses background clutters well and generates visually good contour maps. Based on the saliency maps compared with other models, this paper makes a few basic observations:

The IT, NP, IS and BL models find it difficult to suppress the fog background. The map did not highlight salient objects but detected the fog background together. It is treated with fog as the foreground. As can be seen from Figure 4, there has a very poor effect.

Saliency maps of the MR, BSCA and GP models show that the fog background areas are too bright, and background and foreground are marked as salient regions at the same time. Therefore, saliency maps are blurred. However, the results are relatively better than IT, NP, IS and BL.

The FT, LR, PD, SC, CA models detect salient objects in the foreground while also clearly detecting non-salient objects such as trees, streetlights, and roads in the back-ground. Such an algorithm cannot achieve the purpose of saliency detection and is meaningless for tracking humans in the wild.

Although the fog background has less interferential in the SR than others, the salient objects are also not detected. It is the worst model on test dataset. Due to the features they used are ineffective in foggy images.

The SO, SMD and MIL have poor performance for foggy images with a slightly complex background. Although salient objects in the foreground are detected, the brightness of the fog in the image affects the final detection results. In other words, these models are not robust in fog environment.

The experiment results show that other models cannot detect the salient objects well under foggy weather. It is evident that the proposed method can better detect the salient objects in foggy images and more effective than other models. The reasons are summarized as follows:
(1)The local and global information of the images are utilized, so that the salient objects can complement the information in the FD and the SD. However, the traditional machine learning method leads to the loss of edge information. Thus, causes the boundary of the saliency map to be blurred.(2)The object contour detection method of deep learning is added to enrich the edge information of the saliency map. It can suppress the interference from the fog background and acquire the edge of salient objects more precisely.(3)In the meantime, traditional based method and deep learning-based method are combined to effectively detect salient objects. The proposed method can not only retain the edge more accurately via object contour detection, but also ensure the salient objects’ integrity. By this means, can obtain a more precise and clearer saliency map.

## 5. Conclusions

In our study, we present a high-efficiency model to handle the salient object detection of foggy images. The proposed model combines traditional machine learning based frequency-spatial saliency detection algorithm and deep learning-based object contour detection algorithm to cope with the matter of salient object detection under fog environments. In traditional saliency detection method, the saliency map is acquired by fusing the frequency and spatial saliency maps via DSWT. Then, a fully convolutional encoder–decoder model is utilized to improve the contour of the salient objects. Experimental results on foggy image dataset demonstrate that the proposed saliency detection model performs obviously better against other 16 well-known models.

## Figures and Tables

**Figure 1 entropy-21-00374-f001:**
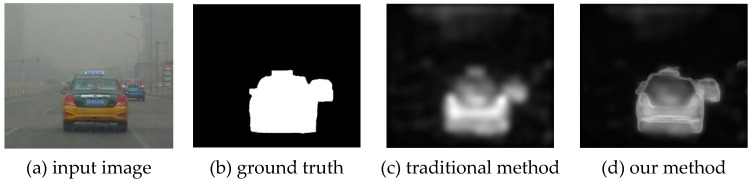
Example of salient object detection in foggy images.

**Figure 2 entropy-21-00374-f002:**
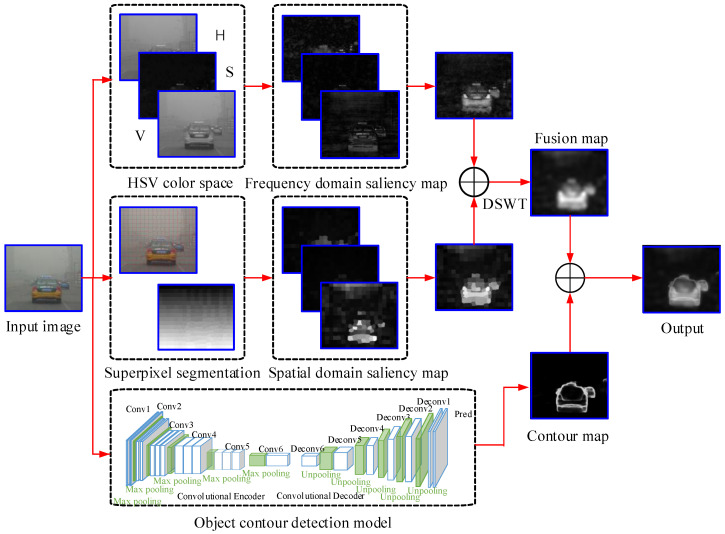
Flowchart of the proposed salient object detection model in single foggy image.

**Figure 3 entropy-21-00374-f003:**
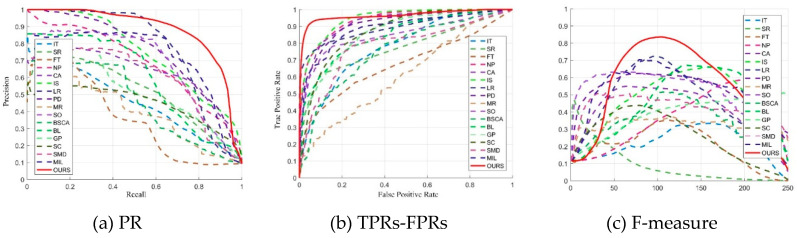
The quantitative comparisons of the proposed saliency model with 16 state-of-the-art models in foggy images.

**Figure 4 entropy-21-00374-f004:**
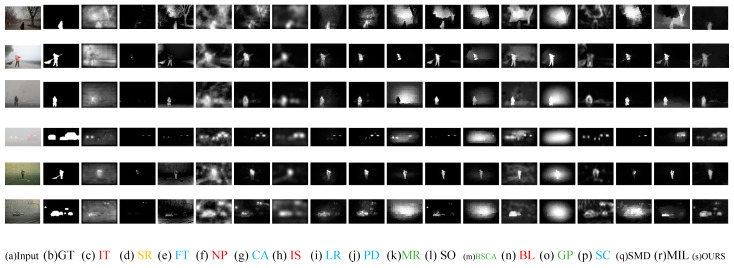
The saliency maps of the proposed model in comparison with 16 models in foggy images. (**a**) testing foggy images, (**b**) ground truth binary masks, (**c**–**r**) saliency maps obtained by 16 state-of-the-art saliency models, (**s**) saliency maps obtained by the proposed model.

**Table 1 entropy-21-00374-t001:** The performance comparisons of various saliency models in foggy images.

Saliency Models	AUC	MAE	WF	OR	TIME(s)
IT	0.7916	0.3434	0.1250	0.1629	5.7661
SR	0.5602	0.1118	0.0730	0.3253	10.7458
FT	0.6809	0.1724	0.1268	0.1703	0.8717
NP	***0.9156***	0.2881	0.1879	0.4357	4.6347
CA	0.8718	0.1328	0.2729	0.4145	59.2286
IS	0.9077	0.1736	0.2378	0.4115	1.2987
LR	0.8687	0.1174	0.2661	0.4274	146.0651
PD	0.8277	0.1073	0.3602	0.4449	28.5625
GBMR	0.5658	0.2219	0.2058	0.1809	2.4929
SO	0.7705	**0.0913**	**0.4431**	0.4557	2.5251
BSCA	0.7327	0.1850	0.2028	0.2420	6.7254
BL	0.8053	0.2220	0.2159	0.5007	53.3103
GP	0.8241	0.2235	0.2556	0.3133	20.2649
SC	0.8077	0.1491	0.2005	0.3215	39.0530
SMD	0.7311	0.1418	0.2976	0.3455	7.6693
MIL	0.8636	0.1365	0.3127	***0.5009***	341.9829
Proposed	**0.9177**	***0.0995***	***0.4060***	**0.6050**	5.0756

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
