# Peer review of "Saliency Detection Based on the Combination of High-Level Knowledge and Low-Level Cues in Foggy Images"

_entropy, 2019, doi:10.3390/e21040374_

Round 1
Reviewer 1 Report
This paper proposes a model to handle salient object detection of foggy images.
The manuscript is well-written and but requires some modifications before publication.
Equations 4, 5, 6, 10, 11, 12, 13, 18, and 22 seem to use MATLAB functions in the equation. Please explain them in mathematical form instead so that the equations are program language-neutral.
Author Response
Question 1:
Equations 4, 5, 6, 10, 11, 12, 13, 18, and 22 seem to use MATLAB functions in the equation. Please explain them in mathematical form instead so that the equations are program language-neutral.
Reply to question 1:
Thanks for the reviewer’s reminder. We have modified and rewritten the corresponding sentences in this manuscript to improve the readability. We have further analyzed these equations in mathematical form and what the parameters mean. The corresponding explanations and supplements have been provided in the revised manuscript. For convenience to the reviewer, the detailed explanations and corresponding modified sentences are shown in attachment file.
Finally, we wish to say that we really appreciate for the reviewers’ useful and valuable suggestions. They are very constructive for improving the quality of the paper.
The manuscript has been resubmitted to Entropy. Thank you very much!
Sincerely,
Xin Zhu, Xin Xu, Nan Mu, Jing Liu, and Heng He

Reviewer 2 Report
My comments and questions are given in the attached file.

Author Response
Dear Reviewer,
We have thoroughly revised the entire paper by incorporating all comments raised from reviewers. Therefore, we submit this revised manuscript, and our replies to each question that is raised from reviewer, for your evaluation. Thanks for your time and effort on organizing the review of our submitted manuscript.
Reviewer #2: In this manuscript, a method for salient object detection of foggy images is presented. A frequency-spatial saliency detection algorithm is combined with contour extraction based on deep learning. Here are my comments and questions:
Question 1:
I am not qualified to judge the English language and style but I feel that some minor spell check is required. I have found the following mistakes:
a. line 42: the word "research" should be probably replaced by "reach",
b. line 55 the verb "is" should be probably removed from this sentence,
c. line 127: one of the words "vertical" should be probably replaced by "horizontal",
If you agree with me please correct these mistakes and check the whole of the paper.
Reply to question 1:
Thanks for the reviewer’s reminder. We have modified the spelling mistakes in corresponding sentences according to your opinion. Then, we checked the whole of the paper and made modifications to other places. For convenience to the reviewer, the related sentences are shown as follow.
Modified sentences:
a. So far, all kinds of defogging techniques [4-7] have been proposed, and they can reach comparatively good performance.
b. (1) High complexity makes their execution time-consuming, thereby making it difficult to guarantee the real-time performance of saliency detection.
c. Sobel is usually directional and can detect only vertical or horizontal edges or both
Question 2:
The proposed method operates in HSV color space. According to the text (lines 202 -204) this color space "...has shown strong stimuli to human visual cortex in foggy images...". Can you support this statement by referring to the literature or to the results of your own experiments?
Reply to question 2:
Thanks for the reviewer’s suggestion. HSV uses three basic attributes of color: hue saturation and brightness to represent color. It is a color model for visual perception. HSV color space can better reflect people's perception and identification of color. It is capable of emphasizing human visual perception in hues and has an easily invertible transform from RGB. We have added relevant reference to support this statement. For convenience to the reviewer, the modified sentences are shown as follow.
Modified sentences:
Given a foggy image, it is transformed into HSV color space firstly, which has shown strong stimuli to human visual cortex in foggy image [40], thus the hue, saturation, value features of H, S, and V channels can be considered as the important indicators for detecting saliency.
[40] Chen, W.; Shi, Y. Q.; Xuan, G. Identifying Computer Graphics using HSV Color Model and Statistical Moments of Characteristic Functions. In Proceedings of 2007 IEEE International Conference on Multimedia and Expo, Beijing, China, 2-5 July 2007, pp. 1123–1126.
Question 3:
According to line 287, the saliency map is thresholded. How the threshold value was selected? Does the method have any parameters? If yes, how are they selected?
Reply to question 3:
Thanks for the reviewer’s reminder. is the parameter to weigh the precision and recall, which is set to 0.3 in the experiment results part. Since the precision is much more important than recall [17], [26], we set =0.3 to emphasize the precision. We have added the threshold value and several relevant references in this manuscript. For convenience to the reviewer, the modified sentences are shown as follow.
Modified sentences:
The precision, recall, TPR and FPR values are generated by converting the saliency map into binary map via thresholding to compare the difference of each pixel with ground truth. is the parameter to weigh the precision and recall, which is set to 0.3 in our experiments [18], [22].
[18] Achanta, R.; Hemami, S.; Estrada, F.; Susstrunk, S. Frequency-tuned salient region detection. In Proceedings of 2009 IEEE Conference on Computer Vision and Pattern Recognition, Miami, USA, 20-25 June 2009, pp. 1597–1604.
[22] Cheng, M.-M. N.; Mitra, J.; Huang, X.; Torr, P. H. S.; Hu, S.-M. Global contrast based salient region detection. IEEE Transactions on Pattern Analysis and Machine Intelligence, 2015, 37, 569–582.
Question 4:
This is the most important one. The experiments have been carried out using two datasets (lines 274-275). Please indicate which images were used for training the contour detection network and which ones were used for testing. Which dataset was used to generate the results shown in Table 1, and figures 3 and 4?
Reply to question 4:
Thanks for the reviewer’s reminder. The experiments have been carried out using two datasets: a foggy image dataset made by ourselves for final test to generate the results shown in Table 1, and figures 3 and 4. And another one is PASCAL val2012 for training the object contour detection model. In addition, ChinaMM18dehaze dataset is our another work for dehazing. However, in this paper we do not train the object contour detection model via this dehaze dataset. We are so sorry that we made a simple mistake here. We have rewritten the corresponding sentences and modified the mistakes in this manuscript. For convenience to the reviewer, the related sentences are shown as follow.
Modified sentences:
A foggy image dataset made by a stand camera, which contains 200 foggy images and the corresponding manual labeled ground truths. The other one is BSDS500 Dataset. It includes 500 natural images with carefully annotated boundaries by different users. The dataset is divided into three parts: 200 for training, 100 for validation and the rest 200 for testing. Object contour detection is utilized to optimize the saliency map which obtained by traditional machine learning methods of salient object detection. Because the saliency map of traditional methods has incomplete edge information.
Question 5:
I am curious how was the comparison with other methods done (lines 309-311)? They have been programmed by the Authors or have the literature results been cited? If the second is true, have individual methods been tested on the same images?
Reply to question 5:
Thanks for the reviewer’s reminder. This manuscript proposed a combination of high-level knowledge and low-level cues based visual saliency detection model in foggy images. To demonstrate the effectiveness of the proposed saliency model, a comparative experiment is presented in Fig. 4 to check the performance and to verify the advantages of our saliency detection method. The source code provided by others was used to test on our dataset. Each foggy image in our foggy dataset was tested on 16 methods of others to produce the corresponding saliency map as shown in Fig. 4. The corresponding explanations and supplements have been provided in the revised manuscript. For convenience to the reviewer, the modified sentences in the experiments are shown as follow.
Modified sentences:
The presented method is compared with 16 well-known saliency detection methods including: IT [15], SR [20], FT [22], NP [45], CA [16], IS [46], LR [47], PD [48], MR [44], SO [49], BSCA [50], BL [51], GP [52], SC [53], SMD [19], and MIL [54]. The source code provided by others was used to test on our foggy dataset. Each foggy image in our dataset was tested on 16 methods of others to produce the corresponding saliency map as shown in Fig. 4.
Finally, we wish to say that we really appreciate for the reviewers’ useful and valuable suggestions. They are very constructive for improving the quality of the paper.
The manuscript has been resubmitted to Entropy. Thank you very much!
Sincerely,
Xin Zhu, Xin Xu, Nan Mu, Jing Liu, and Heng He
Round 2
Reviewer 2 Report
All my comments and questions from the previous round have been addressed. I believe that the manuscript has been significantly improved. I only have one additional question.
The experiments and the comparison have been carried out on the Author's own dataset. Therefore, I think that it is necessary to make this dataset publicly available. In this way, other researchers will have the opportunity to compare their methods with the presented algorithm. It should also have a positive impact on the article's citation. Could you share your dataset?
Author Response
Thanks for the reviewer’s suggestion. The experiments and the comparison have been carried out on our Foggy Images dataset. We uploaded the dataset to the google drive: https://drive.google.com/file/d/1aqro3U2lU8iRylyfJP1WRKxTWrrFzizh/view?usp=sharing.
Modified sentences:
A foggy image dataset (FI) is collected from the Internet, which contains 200 foggy images. We also provide the corresponding manual labeled ground truths. The FI dataset can be downloaded at https://drive.google.com/file/d/1aqro3U2lU8iRylyfJP1WRKxTWrrFzizh/view?usp=sharing. The other one is BSDS500 Dataset. It includes 500 natural images with carefully annotated boundaries by different users. The dataset is divided into three parts: 200 for training, 100 for validation and the rest 200 for testing. Object contour detection is utilized to optimize the saliency map which obtained by traditional machine learning methods of salient object detection. Because the saliency map of traditional methods has incomplete edge information.
